# Gifted Children through the Eyes of Their Parents:Talents, Social-Emotional Challenges, and Educational Strategies from Preschool through Middle School

**DOI:** 10.3390/children10010042

**Published:** 2022-12-25

**Authors:** Roberta Renati, Natale Salvatore Bonfiglio, Martina Dilda, Maria Lidia Mascia, Maria Pietronilla Penna

**Affiliations:** 1Department of Pedagogy, Psychology, Philosophy, University of Cagliari, 09126 Cagliari, Italy; 2Noah Srl, 27100 Pavia, Italy

**Keywords:** gifted children, parenting, educational strategies

## Abstract

Few studies have been conducted analyzing the experience of raising a gifted child. The present exploratory study focused on examining the profiles of a sample of 44 gifted children, exploring aspects related to health status, precociousness of development, and peculiarities of their potentiality and peculiar emotional profile. Through the administration of a semi-structured questionnaire and an in-depth interview, the experience of parents of gifted children was also analyzed, deepening the challenges they have to face and the educational strategies they implement. The evidence that emerged helps shed light on specific aspects that characterize gifted children and have implications for family educational practices.

## 1. Introduction

Trying to provide a framework regarding the profiles of gifted children and the challenges faced by their parents is a complex issue, not only because scholars’ views do not always converge but, mainly, because it requires the analysis of different levels of observation. First of all, the complex interplay between individual and contextual factors that, together, contribute to shaping a person’s developmental trajectory [1,2,3,4,5]. 

Recent studies have questioned the traditional conception that considers giftedness as an immutable trait of the person, linked merely to the individual’s cognitive abilities [6]. Indeed, although genetic influences have been found to play a role in the emergence of giftedness [7], many other factors relating to psychological and social dimensions appear to be central to the development and expression of individual potential [6,8]. While IQ is a crucial predictor of academic performance and success, it alone cannot explain intra-individual variability [9,10]. This evidence allows us to reinterpret giftedness from a developmental perspective, which considers the complex interdependence of different variables at the individual, familial, educational, and societal levels [11,12,13]. Concerning this perspective that considers the expression and development of potential along the lifespan, the possibility of intervening at an early stage becomes crucial to create the environmental conditions that can best support the gifted child’s harmonious development of cognitive and socio-emotional competencies. Thus, there emerges the importance of creating favorable developmental environments that can provide adequate learning opportunities and proper nurturing to foster the development of the so-called life competencies, crucial protective factors to adapt to a changing world [14]. Despite the limitations of research in this area, some studies have emphasized how different outcomes of gifted children depend on the family environment [15,16,17]. Parents play a crucial role in accompanying gifted children in developing the self-regulatory skills necessary to develop their potential and promote their psychosocial well-being. 

This exploratory study aimed to provide new insights into the experience of parents of gifted children by focusing on parents’ views of their children’s cognitive and social-emotional profiles. Specifically, in relation to the different ages of the children involved in the study, we shed light on their specific characteristics, the educational challenges faced by parents, and their educational strategies. The ultimate goal was to provide valuable suggestions for the development of parenting support interventions.

## 2. Characteristics and Profiles of Gifted Children

In the psychological field, gifted children score two standard deviations above average on Intelligence Quotient tests. Within the continuum that defines the distribution of intelligence, giftedness is the psychological reality that accounts for 2.14% of the population (IQ ≥ of 130). This percentage expands to reach approximately 5% to 8% of the population if, as a lens of observation, one uses explanatory models that do not reduce the concept of giftedness to the IQ value alone, but that define giftedness as an above-average cognitive ability (IQ ≥ of 120) that interacts with other individual and contextual variables [10,18]. As giftedness is a very complex and multifaceted issue, there is no single universally accepted definition of “gifted children.” There is, however, a general agreement between scholars that giftedness can be described as a complex set of genetic, psychological, and behavioral characteristics resulting in outstanding abilities [19,20] in one or more areas, such as general intellectual ability, specific academic aptitude, creative thinking, leadership, and visual and performing arts. Pfeiffer (2013) [21] stated that “the gifted child demonstrates a greater likelihood, when compared to other students of the same age, experience and opportunity, to achieve extraordinary accomplishments in one or more culturally valued domains” (p.14), emphasizing how potential can refer and be expressed in any domain of experience that is valuable within the person’s specific cultural context. Linda Silverman [22,23] emphasizes the centrality of the gifted child’s intellectual and emotional experience, stressing that giftedness should be understood as something that involves the whole psychological and behavioral experience of the person. This theoretical view affirms that giftedness is something that the individual simply “is”. The Columbus Group’s definition [24] stresses that giftedness can be defined as asynchronous development in which the subject’s advanced cognitive abilities and exceptional intensity interact with each other, generating inner experiences and awarenesses in the person that are qualitatively different from the norm. This asynchrony becomes more significant as the level of a child’s cognitive potential increases, giving insight on the unique psychological and sensory experiences that these children go through, experiences that often make them particularly sensitive and vulnerable.

The literature on the characteristics of gifted children points out that they can have different profiles [25,26] and different levels of potential [6,27,28,29], highlighting the principle that each gifted individual is unique. Neihart [30] emphasizes the heterogeneity of these kids, pointing out that they can come from all walks of life and nationalities, as well as from all ethnic and socioeconomic backgrounds. These children exhibit peculiarities related to certain temperamental aspects, the precocity and speed with which they learn, the way they cope with challenges and manage moments of difficulty, and the perseverance and passion they invest in achieving the goals they set for themselves. 

When children exhibit high cognitive potential, certain singularities can be observed even at an early age. Generally, these children are precocious in reaching major developmental milestones and quicker in making the progress that will lead them to the subsequent stage. They are extremely active and inquisitive in exploring their closest physical and social contexts, as well as quick in engaging parents and teachers with constant solicitations. Typically, these children learn to read and count early, have a very efficient memory and strong problem-solving skills. These peculiarities lead them to acquire more extensive and in-depth knowledge than their same age peers, as well as the ability to think abstractly, confronting with complex and emotionally activating concepts. It should be underlined that these advanced capacities at the cognitive level are not always complemented by adequate emotional competencies; it may therefore happen that children are not able to manage the emotions elicited by the experiences and content they can access at the cognitive level. The literature explains these facets by the concept of developmental asynchrony, referring to the fact that these children often show different levels of development with respect to the cognitive and emotional domains.

There are no epidemiological data on the prevalence of psychological and social problems in the gifted population; nonetheless, scholars have pointed out that gifted children and adolescents may face the same difficulties as their peers with typical development [31,32], such as bullying and relationship problems, impulse control difficulties, mood disorders, addictive disorders, anger management issues, and family conflict [33]. In addition, gifted children may have a specific learning disability or ADHD [34]. Porter [35] identified some risk factors that seem to characterize gifted and talented children; these include overexcitability, low self-esteem, perfectionism, anxiety and stress management struggles, depression, behavioral and social difficulties. Additional factors that appear to affect gifted child adjustment include asynchronous development [36], underachievement [37], excessive parental engagement, the mismatch between child needs and the educational environment, and difficulties with the peer group [38].

In general, studies that investigated gifted children’s psychological and social adjustment presented mixed results. For example, some researchers found that gifted children are more resilient [30,39,40] and thus more successful in coping with stress. Other studies showed that this population of kids might be psychologically vulnerable [41,42,43,44,45], displaying adjustment and behavioral problems [46,47], especially when they come from a deprived background or an ethnic or cultural minority [48].

## 3. Parents’ Experience in Raising a Gifted Child

The family plays a crucial role in the development and adjustment of the gifted child. Parents are often the first to notice the child’s peculiarities and guide them in the first steps of identifying the specific potential or talent profile. They are dedicated to meeting the child’s intellectual and emotional needs, investing significant time and effort in providing the appropriate stimulation and environment to meet the child’s unique needs. Few scholars have investigated parents’ understanding of their gifted children’s singularity and the challenges they have to deal with in parenting practices [49].

The majority of scientific publications focused on investigating the role of the family in the development of talent, emphasizing how parents can support gifted children in achieving academic success [50,51]. Some studies explored the degree of parents’ satisfaction with school programs [16,52] and their need to understand giftedness [53]. The limited number of studies that have focused on exploring and describing the experience of raising a gifted child found that parents of the gifted have to deal with unique concerns and challenges that can represent a source of stress [49,54,55,56,57]; Meckstroth [58] described the experience of giftedness in the family as a “crisis cubed” that affects all situations in daily life and in which feelings and perceptions are intensified. Some scholars highlighted that parents of the gifted need guidance in parenting practices [16,23,59]. 

Children with high abilities tend to come from cohesive, child-centered families characterized by an authoritative parenting style where parents invest a lot of energy and commitment in their children [31]. Parents of gifted children seem to fully recognize the uniqueness of their children, but only a few understand their child’s ability profile and emotional peculiarities. Therefore, they may struggle to manage the child and meet his needs.

Parents may perceive difficulties in relation to some aspects related to exercising their role and may have low confidence in their ability to manage and support their gifted child [60]. Parents may also feel the responsibility to provide their children with the proper opportunities, investing considerable effort in seeking appropriate accommodations for their children’s unique learning needs, as well as advocating for them in the school setting [61]. Studies have highlighted that parents of the gifted find themselves dealing with the overwhelming urgency of their children’s demands, including high levels of emotional intensity. In these situations, parents are inclined to implement negotiation strategies and to be accommodating, in order to avoid tiring, intense emotional outbursts of the child, and moments of high levels of conflict [54]. Parents also play a crucial role with regard to their gifted kid’s emotional management [31,62]. In addition, it has been pointed out that these parents may feel unsupported [63] and experience a marked sense of loneliness [57,64]. Due to misconceptions about the nature and manifestations of giftedness, they may feel misunderstood even by friends and family members. The profound differences between the gifted child and the same age non-gifted peers may also cause parents to experience feelings of guilt and shame [56]. There is a specific need for counselling programs for families of gifted children, but this is still an under-explored area that needs to be implemented.

It is crucial to clarify that having one or more gifted children does not necessarily imply having difficulties in family life; however, parents ask for support on some issues for which they feel unprepared [65]. In particular, parents ask to be guided so their children can be confident with themselves and others, blossoming at the human level and fulfilling their potential.

### Counselling Programs for Gifted Families

It is recognized that there is a need to implement family interventions for parents of gifted children, focused both on those situations in which children manifest problems as well as in the area of prevention, targeting children identified as gifted but who currently do not display difficulties [66]. 

Few intervention programs have been developed to address the parenting needs of this specific population of parents and are mostly characterized by insufficient evidence of effectiveness [67,68,69]. Numerous parent counselling interventions are validated in terms of effectiveness, but, to the authors’ knowledge, these programs are rarely tailored to meet the unique needs of parents of gifted children. Morawska and Sanders [70] have adapted the Triple P Positive Parenting Program [71] to meet the needs of parents of gifted children by tailoring and testing the Gifted and Talented Triple P program [59,70]. The program involves teaching parents fundamental child management skills, divided into three main domains: (1) promoting child development; (2) managing misbehavior; and (3) planning activities and routines. Some specific aspects of parenting the gifted are emphasized, such as: having clear expectations of children, problem-solving skills, promoting children’s self-esteem, encouraging persistence and perseverance, having effective rules and boundaries, helping children establish good relationships with siblings and peers, managing anxiety and other emotions, and building a good school–home alliance. Participants in the intervention condition showed statistically significant improvements and clinically reliable changes; parents reported a reduced number of their child’s problem behaviors and a lower frequency of challenging behaviors following the intervention. However, there was no effect on the child’s emotional symptoms or difficulties with peers [70]. A recent study [72] highlighted the effectiveness of psychoeducational and systemic intervention in improving parental awareness of gifted children. These findings are consistent with previous studies [73,74,75] and support the idea that systemic-oriented interventions are valuable. Such programs can promote new perspectives in the family [74,76] and enhance parents’ awareness and understanding of the nature of giftedness [77].

In light of the above-mentioned findings, the present study aims to deepen the understanding of some aspects related to the profiles of gifted children that may have an impact on parenting and investigate parenting challenges and strategies by investigating the following research questions.

RQ1: Which kind of health-related issues characterize gifted children? 

RQ2: What are the areas of potential and the emotional peculiarities that parents identify in their children with respect to different ages and levels of giftedness?

RQ3: What educational challenges do parents feel they face in relation to the different ages of their gifted children?

RQ4: What strategies do parents implement in relation to the ages of the children?

## 4. Materials and Methods

Parents were given oral and written information about the purpose of the study and signed informed consent and privacy forms, including for the child’s evaluation. Parents were asked to fill out together an open-ended questionnaire designed to collect anamnestic information about the child’s health status and development milestones; the survey form included a section designed to gather the parents’ views on the child’s cognitive and emotional profile, as well as a description of the areas of challenge they were facing and the educational strategies they were implementing. Once the questionnaire was completed, an in-depth interview lasting about 1 h and 30 min was conducted with each parental couple, aimed at exploring more deeply their answers to the questionnaire and clarifying possible interpretive doubts. To investigate the research questions, a mixed method combining quantitative and qualitative data collection and analysis was adopted. Regarding the quantitative analyses, before conducting the analyses, assumptions of normality were performed for the dependent variables using the Levene statistic for homogeneity of variance and visual analysis of the Q-Q plot, Skewness, and Kurtosis. All indices resulted in the normal range. A Chi-square analysis was performed for variables related to health problems and developmental precocity, in relation to levels of giftedness and age groups. A univariate ANOVA was carried out considering the WISC-IV indices as independent variables as a function of children’s age. Absolute frequencies were calculated for variables related to children’s areas of potentiality, social-emotional characteristics, parents’ educational challenges, and educational strategies. With respect to the qualitative data, an inductive approach was taken to answer the research questions [78]. Two independent researchers conducted an in-depth reading and analysis of the parents’ responses to the questionnaire; the content was appropriately coded to identify information related to the questionnaire areas, identifying themes related to children’s profiles, parenting challenges, and educational practices. We categorized the responses according to content categories, using a data-driven approach and an open coding procedure without imposing a predetermined set of categories on the data. During this process, content that was too vague and ambiguous was discarded. The analysis included three rounds of review that resulted in a categorization of all responses for each of the 4 areas investigated. 

### 4.1. Participants

Participants constituted a community sample. Forty-four families of gifted children were recruited on a voluntary basis and without honorarium from an Italian center specializing in gifted assessment. All parents were explained the purpose of the study and provided written informed consent. Eighty-seven parents participated in the study, 43 mothers with a mean age of 42 years (SD = 4.6) and 44 fathers with a mean age of 45.2 years (SD = 6.7). Most of the couples were married and lived in northern Italy. The majority of the parents were well-educated, holding a university degree; see Table 1 for parents’ educational qualifications. All the parents declared to be employed, and the family SES was middle or upper class. 

The participating children were 44, aged from 5 years and 6 months to 14 years (mean age = 8.3; SD = 2.47) participated in the study. The children were divided by age groups: 4 children were 5 years old, 16 children 6–7 years old, 16 children 8–10 years old, and 8 children 11–14 years old. All the children received a formal comprehensive assessment; intellectual ability was tested using the WISC-IV, the full-scale IQ score ranged from 121 to 156 with a mean of 135.7 (SD = 8.30); average IQ full-scale scores sorted by age group are provided in Table 2. With respect to the level of giftedness, children were distributed into 3 categories: mildly gifted with FSIQ between 120 and 129 (N = 12), moderately gifted with FSIQ between 130 and 145 (N = 27), and 5 highly gifted with FSIQ > 146 (N = 5). All the children involved in the study had no reported history of significant behavioural or emotional problems.

### 4.2. Instruments

***Informative questionnaire.*** The semi-structured questionnaire was designed to meet the purposes of the research and was divided into two sections. The first section was organized with closed multiple-choice questions aimed to collect socio-demographic information about the parents and the child, data on the child’s achievement of major developmental milestones, as well as information about the child’s general health status and the existence of any diagnoses. The second section of the questionnaire consisted of open-ended questions in which parents were asked to describe aspects of the child’s profile and parenting practices. Specifically, parents were asked to describe any areas of potential they found in their child and the main issues related to emotional functioning; concerning parenting practices, parents were asked to describe the challenges they were facing at that particular stage of the child’s development and the educational strategies they were implementing. 

***WISC-IV Wechsler Intelligence Scale for Children- Fourth Edition*** [79]. The Wechsler Intelligence Scale for Children-IV (WISC-IV) is a general intelligence test consisting of 15 subtests, 10 core tests, and 5 supplemental. The 10 mandatory subtests contribute to the measurement of four index scores: the Verbal Comprehension Index (VCI), Perceptual Reasoning Index (PRI), Working Memory Index (WMI), and Processing Speed Index (PSI). The full-scale IQ score FSIQ is computed from 10 main subtests (three related to Verbal Comprehension, three to Perceptual Comprehension, three to Perceptual Reasoning, two to Working Memory, and two to Processing Speed) that reflect the general cognitive ability. The scale can be administered to children and adolescents between 6 years and 16 years and 11 months. 

## 5. Results and Discussion

### 5.1. Health-Associated Characteristics 

In contrast to the myth of intelligence being linked to a predisposition to physical weakness and health problems, the few scholars who have been investigating health issues in gifted children have pointed out that children characterized by high levels of intelligence are active, social, and mentally healthy [80]. Parents were asked if their children had any particular health issues; in particular, the presence of intolerances/allergies, skin problems, sleep problems, and food selectivity were investigated. Results, detailed with respect to both the level of giftedness and the age of the children, are summarized in Table 3.

Regarding allergies and skin problems, research findings highlights that people with cognitive overexcitability tend toward central nervous system hyperresponsiveness [81], which can lead to various other psychological and physiological consequences. Some studies indicated the combination of high intelligence (particularly verbal abilities) with various allergies and asthma occurring from early childhood [82,83,84,85,86]. It appears that the degree to which symptoms of some types of allergies are manifested is associated with physiological and cognitive factors [87]. Some studies have shown a greater occurrence of allergies in intellectually or mathematically gifted children [88]. Higher allergy tendencies have been associated with higher regional structures gray matter volume in areas related to higher-order cognition and better spatial abilities [87]. In our sample, issues related to allergies and skin problems affect a minority of children compared to all levels of giftedness and age groups. 

Both sleep problems and food selectivity can be extremely frustrating behaviors that are not easily manageable by a parent. Not sleeping a lot may be common for some gifted children characterized by high levels of emotional intensity. The little research in this area is based on small cohorts of subjects and shows mixed results. A recent study [89] found no significant differences between gifted and non-gifted children in the prevalence of sleep problems at any age. The same result was found in the study by Piro et al. [90] in which no significant differences were found in average bedtime, hours slept, sleep problems, or use of electronic devices before bedtime. In contrast to the above-cited studies, the report by Bastien and collaborators [91] showed that having a gifted profile increases the risk of having sleep problems by 4.67 times, which appears to negatively impact a child’s adjustment, undermining the child’s emotional and behavioral functioning. With regard to our study, there is evidence of a low percentage of gifted children manifesting sleep problems. During the interviews, some parents reported the presence of pavor nocturnus episodes mainly related to times of change (e.g., transfers, transitions from one school cycle to another) and developmental transitions regarding the child’s ability to understand and deal with emotionally abstract and complex issues.

Food selectivity describes a child’s tendency to have a limited food repertoire and show aversion to certain tastes, textures, colors, types, or brands of food, resulting in restriction of the variety of food intake. Often there is evidence of high-frequency consumption of a single food. This factor appears to be the most significant, as it is present in half of the children in the sample, with respect to all levels of giftedness and at all ages. In addition, parents report a tendency for their children to eat only certain foods and difficulties in eating foods with particular kinds of textures.

### 5.2. Sign of Giftedness: Developmental Precocity, Areas of Potentiality, and Emotional Peculiarities

From a developmental perspective, gifted children often reach “milestones” earlier than their intellectually average peers. Each child’s development is unique; however, gifted children present developmental differences from average children in cognitive, language, social-emotional, and physical domains [92,93]. Gifted children usually have extraordinary memory [94]. Furthermore, from the earliest months of life, these children are characterized by specific cognitive characteristics, such as attention, curiosity, and the need to constantly interact with the environment [35]. Most gifted children start speaking earlier than others and tend to have a more advanced vocabulary than their peers. Some gifted children may start speaking later but when they begin talking, they display an unusually broad and complex vocabulary [61]. Also at a very early age, gifted children are able to appreciate the nuances that distinguish words and to understand abstract concepts. Their sophisticated verbal skills usually lead them to be precocious readers, and they often read widely. Even before they can read, they are motivated to autonomously learn letters and numbers. Not all gifted children exhibit these extraordinarily high verbal skills; for example, mathematically gifted children, particularly males, may not have high verbal skills [95]. This is also true for children who exhibit artistic, mechanical, or athletic talent. In addition to early and extensive vocabulary development, other characteristics of gifted children that are frequently cited in the international literature are sustained long attention span, excellent memory, curiosity, early reading ability, learning speed, and ability to generalize concepts [96,97], excellent problem-solving skills [98], extensive use of abstract thinking, and vivid imagination [99]. Other studies have revealed additional characteristics such as high activity levels, lower need for sleep, ambidexterity, imaginary companions, allergies, sense of humor, sensitivity, perfectionism, focus on morality and justice, and predilection to relate to older children and adults [98,100]. In light of this evidence, in the following paragraphs, the main findings based on the research questions are outlined.

With reference to the developmental precocities investigated, it emerges that language precocity characterizes the majority of gifted children in our sample. On the other hand, motor precocity is reported by a very small number of parents and therefore seems not to characterize the children of our sample. Regarding mathematical precocity, defined as the ability to perform mathematical operations and to think in mathematical terms, a significant difference emerges that indicates this trait is specific to the children in our sample. The same result applies to precocity in writing (see Table 4).

To facilitate the data description and reading, the characteristics described by parents were coded and categorized within the main areas of potential expression, such as general intellectual ability, specific academic aptitude, creative thinking, socio-emotional skills and leadership, visual and performing arts, and athletic ability; to these, the area related to sense of humor was added because it was indicated as an area of children potentiality by some parents. For each area, the peculiarities that emerged in relation to the age of the children were analyzed (see Table 5).

In relation to the *general intellectual ability* area, parents of preschool children identify certain peculiarities such as early and marked verbal skills, autonomy in the acquisition of reading and writing literacy, learning speed, an excellent memory, curiosity, and specialized interests. Parents of 6- to 7-year-old children, in addition to the specificities highlighted by parents of younger children, point to the presence of keen logical reasoning skills, marked attentional skills, and the presence of interests in contents and activities not typical for the age. For example, in the descriptions of parents of 8- to 10-year-olds, traits such as reasoning skills, learning speed, curiosity, and specific interests return, adding unusual observational ability. Parents of preadolescent children report abstract reasoning skills, a good memory, learning speed, curiosity, and specialized interests as distinctive potentiality hallmarks of their children. Referring to the specific academic aptitude area, parents of preschool gifted children did not identify any particular elements, while parents of 6–7-year-olds highlighted math skills, to which abilities in the computer area are added for children 8 years and older. *Creativity* is mentioned by all parents and seems to be a distinctive trait of gifted children. Some parents, while describing this characteristic, emphasized their children’s drive to develop projects and inventions, often related to solving relevant environmental and public health problems. Concerning the *socio-emotional and leadership* area, the parents of children age 5–7 years emphasize the presence of high emotional sensitivity, and in the case of children age 8 and older, empathy is highlighted in addition to sensitivity. With regard to the area of *visual and performing arts*, in our sample of gifted children, these peculiarities are reported by parents from the age of 6 years; particularly, parents of children aged 6 to 10 years mention the area of music, and parents of children aged 11 years and older cite artistic abilities. Regarding abilities in the athletic area, there are only two gifted children whose parents highlight this specific aptitude, and they belong to the preadolescent group. In general, the families interviewed describe the physical ability of their children as “underdeveloped” or “deficitary,” emphasizing aspects related to motor clumsiness, laziness, but also of lack of interest for sports. Some parents express concerns about their children’s disinterests and difficulties in participating in team sports activities. Some fathers perceive this peculiarity as a limitation for the child’s social development. Finally, some parents of children aged 8 and older point out the presence of a strong *sense of humor*.

Part of the characteristics that define the gifted profile, such as high levels of sensitivity and a heightened sense of morality, may make these children more vulnerable to stressors [101]. These children may have very intense emotional reactions to events and situations that are frustrating to them. For example, they may often feel involved and concerned about perceived injustices and social problems that do not typically affect kids, such as poverty, war, and violence. While they are able to capture the significance of specific topics, even existential ones such as “death”, they do not always have the resources to handle the related emotional implications. Therefore, the acquisition of appropriate emotional regulation skills becomes essential. Another aspect that needs to be properly monitored concerns the issue of perfectionism, expecially if it is associated with high expectations regarding one’s performance, as it can lead to the development of anxious symptomatology and avoidance of learning proposals and social relationships. Regarding the social domain, children’s high level of competence and the presence of uncommon interests compared to the chronological age non-gifted peers can in some cases constitute a risk factor for relationships, as these students lack a space for sharing and communication with peers. The gifted child may experience deep sentiments of inadequacy and loneliness, feeling “different” from peers who do not understand or find a connection with the gifted child’s interests and passions.

Regarding the results of our study, the emotional profile described by parents of preschool children is characterized by high levels of emotional sensitivity and empathy that make children responsive and caring about the emotional world and experience of others. 

Parents also indicate the presence of emotional regulation difficulties in children when confronted with obstacles and setbacks. Parents of a five-year-old reported: “*He can’t handle frustration; he goes into crisis in the face of denial and setbacks*”. Some parents report episodes of high emotional intensity in which outbursts of anger may be present. Parents of a four-year-old talking about their gifted child said: “*He experiences emotions intensely and has frequent moments of anger, especially when adults or children do not understand him quickly. He is very empathetic and has a particular respect for both nature (which he recognizes as a source of life and death) and animals*”. These situations are not easy to manage for parents, especially when they occur in public situations. 

In addition, parents also refer to the presence of oppositional behaviors, especially when the child has to manage the unexpected or the imposition of rules. One parent couple reports the presence of “*perfectionism and anxiety related to performance*”. Characteristics that emerge with respect to the emotional profile of 6- to 10-year-olds follow those of younger children, although parents also report a tendency to enact internalizing behaviors to cope with stress, such as emotional closure. Parents of a nine-year-old boy stated: “*He tends to internalize stress and emotions in such situations you have to approach him calmly and make him slowly open up. At that point, he calms down. In the case of positive emotions, however, he tends to hold them back and not fully manifest his joy as if he has some kind of shame*”.

In general, fewer episodes of anger are reported in children in the 8–10-year-old range. A couple reported the presence of sensory hypersensitivity and, another, occasions when the child displays sadness. Similarly, to dealing with emotional outbursts, these internalizing behaviors are described by parents as a source of distress. Parents of a ten-year-old child said *“He lives and sometimes manifests emotions in a very intense way. The suffering of others is his suffering.” He suffers from “world hunger”, which he would like to eradicate when he grows up. If he sees a needy person, he feels guilty about their suffering. His reactions are often instinctive, and he adapts with difficulty to frustrations. He does not like the unexpected”.* Furthermore, parents describe the emergence of a deep sense of justice that can be a stressor causing episodes during which their child exhibits oppositional behaviors. In general, the difficulty in emotional regulation appears evident, which can lead to the onset of conflict episodes in the family, at school, and with peers. Regarding the group of 11- to 14-year-old children, parents point out that, in addition to sensitivity and empathy, low self-esteem, anxiety, and worries emerge. The mother of a thirteen-year-old girl said: *“She’s very anxious; she struggles to get out of her comfort zone”.* This, in some situations, results in a propensity to self-enclosure and a tendency to exhibit low mood. A father of a twelve-year-old girl reported: *“She is particularly sensitive, having outbursts of anger that she alternates with episodes of extreme sadness”.* Even if to a minor extent, some difficulties in managing emotions seem to persist, which may also be associated with the significant biological changes imposed by the pubertal spurt. In general, parents report a concern about the risk of social isolation. Main results are showed in Table 6. 

### 5.3. Parenting: Challenges and Educational Strategies

The central challenges reported by parents of preschoolers relate to managing emotional intensity, especially when the child is struggling with frustration. Some parents express difficulty in comprehending their child’s unique characteristics. Regarding the educational strategies adopted, parents of younger children refer the necessity of being open to giving emotional support to the child, as well as receptive in listening to the child’s needs, in order to understand the causes of certain behaviours. Dialogue, detailed and consistent explanations, also supported by readings, and negotiation are the most frequently adopted strategies. The main challenges reported by parents of children aged 6 to 10 concern the management of emotional and behavioural aspects (e.g., “emotional immaturity”, “acceptance of rules”). More specifically, parents report issues related to the school area and socialization emerge. Regarding the school area, parents report difficulties related to managing the child’s feelings of boredom and low levels of motivation, as well as problems associated with the lack of dialogue and educational alliance with teachers. Parents mention school as a significant source of stress both for them and the child. Regarding socialization, concern emerges about children’s difficulty in building relationships with peers because of their different interests. For parents in the preadolescent group, managing school difficulties seems to become increasingly significant and is associated with handling their children’s feelings of low self-esteem and social isolation. Parents also refer to problems in managing the use of technological tools that are used by their kids to both find new stimuli and socialize. The issue of defining one’s own identity also emerges; parents report preoccupation about the distress their children show regarding their social world. The educational strategies implemented by parents of children between the ages of 6 and 14 differ only partially from those implemented by parents of younger children. Strategies such as emotional availability and open dialogue persist for all ages. These are complemented by advocacy intervention carried out with the child’s teacher, promotion of activities and experiences related to the child’s specific interests, and encouragement of socialization. Also emerging is the recourse to specialists in the field of giftedness who can support both the parents in their educational role and the children in the understanding of certain aspects of themselves. The details about the educational challengies and strategies reported by parents are shown in Table 7 and Table 8.

## 6. Summary and Conclusions

Research indicates that the parenting experience of gifted children’s parents differs from those of children who do not have a gifted profile [23,49,55,56,57,60]. The results of the present study contribute to enrich the literature on gifted children’s profiles and the challenges faced by their parents.

The study also aimed at exploring gifted children’s health issues, a topic rarely addressed in the literature. The results indicate that only a small group of the children in our sample presented issues related to allergies, skin problems, and sleep disturbances. In contrast, half of the children showed food selectivity, particularly concerning difficulties in eating certain types of foods and tolerating their consistency. Precocities in the acquisition of major developmental milestones were shown, particularly concerning language and reading–writing acquisition; precocity was also shown in the logical-mathematical area. Finally, with reference to the area of motor development, almost all children in our sample did not show any precocity. 

In relation to the areas of potentiality that parents identify in their children, distinctive traits that are in line with the international literature emerged. Characteristics related to creative thinking and specific academic aptitudes were highlighted, particularly concerning mathematics and computer science topics. Regarding emotional aspects, the presence of deep sensitivity and empathic competencies emerged. Referring to the area of artistic potential, parents primarily reported specificities related to musical talent. Athletic aptitude was cited as an area of strength only for two gifted children in the sample, and some parents reported humor as a typical characteristic of their children. Traits such as speed of learning, advanced verbal skills, sharp memory, intense curiosity, uncommon interests for their age, and observational skills emerged and were reported as traits related to the general intellectual ability area. 

The emotional profile of gifted children that emerged from this study is characterized by emotional sensitivity, empathy, and a sense of justice, but also by marked difficulties in managing emotions and the presence of oppositional behaviors that arise in relation to handling frustration. These aspects can add complexity to parenting. As age increased, emotional traits linked to negative emotions concerning topics such as self-image and relationships with peers emerged; furthermore, issues related to anxiety and worry, which may be linked to traits of perfectionism, were also highlighted. Regarding educational challenges, parents mainly reported children’s difficulties in managing emotions, which can be related to the typical developmental asynchrony that distinguishes gifted children. The school context emerged as a significant source of stress for parents, primarily because of concerns about social difficulties they seem to perceive in their children, related mainly to the possibility of bonding with their peers. The educational strategies that parents reported implementing were associated with to an educative approach characterized by listening, emotional openness, and dialogue, with reference to all age groups considered. In relation to children’s entrance into the school system, parents reported the necessity to implement strategies to make the school aware of their children’s learning and social-emotional needs, serving as a sort of advocate for them. 

Our findings are helpful for parents of gifted kids and provide insights for both research and clinical practice. Parents can identify some factors that may affect the experience of parenting a gifted child, stimulating them to self-reflect on their educational approach. In addition, knowing that other parents are facing specific challenges may make them feel less alone. Regarding clinical practice, mental health professionals need to be aware of the characteristics of giftedness to recognize what is typical and atypical for this population. Specific traits, such as intensity, sensitivity, and perfectionism, can be seen as dysfunctional by a professional not trained in working with gifted children, also leading to the risk of misdiagnosis. Parents need counselling because they require support in dealing with unique issues such as heightened intensity, perfectionism, or problems establishing social relationships with peers [66,102]. Parents can also be concerned about identification, labelling, and placement. In particular, our study highlighted distinctive elements concerning gifted children’s socio-emotional needs. 

Scholars have highlighted the substantial role of families as a “context” that can enhance processes that promote positive outcomes in children in the face of daily stresses and difficulties [103]. The ecosystem in which a child is embedded is one of the most critical factors in promoting positive developmental outcomes and resilience, so the trajectory leading to a child’s well-being can be affected, positively or negatively, by the dynamic interplay between individual, family, and environmental factors. If we consider the family as a complex system, the special needs of the gifted child can be a unique source of stress for parents and siblings, especially when asynchronous development is substantial. Therefore, parenting practices are crucial issues to explore in the gifted field, where research on parenting experience is lacking [53,59,60,70]. The scientific literature points to the need to develop intervention programs for parents of gifted children [60,72,104,105,106]. In light of the evidence from this study, intervention programs should focus on parents’ comprehension of giftedness and the identification of concrete educational strategies helpful in supporting gifted children who experience transitory social-emotional difficulties related to their unique profile. Counselling interventions for parents should focus on educational approaches that are useful in reducing parental stress levels and helping parents become more confident in their role. At the psycho-educational and clinical level, it is essential to develop evidence-based interventions with empirical evidence, such as the triple P program, which has also been successfully applied to families of gifted children [70]. 

While there is an emerging need to develop supportive programs for the families of gifted children, there is also a need to reflect on mental health professionals’ training regarding the characteristics of gifted children and the necessities of their families.

The present study has some limitations that need to be considered. First, voluntary recruitment could incorporate selection bias, having reached only the most willing and motivated parents. In addition, there emerges a need to expand the sample to explore gender differences, including across developmental stages. In addition, future research should also examine the experience of siblings and use a between-groups research design.

## Figures and Tables

**Table 1 children-10-00042-t001:** Parents’ educational qualification and family residency.

	Level	Counts	Total
Education (mother)	Secondary School	13	44
	University bachelor	18	44
	University master	9	44
	Ph.D.	4	44
Education (father)	Secondary school	22	44
	University bachelor	17	44
	University master	5	44
Family area of residence	Nord	39	44
	Sud	4	44
	Islands	1	44

**Table 2 children-10-00042-t002:** WISC IV indexes average scores and standard deviations categorized by children’s age.

		Age Categories					
		<6	6–7	8–10	11–14	F	*p*
**FSIQ**	M	138.8	135.5	134.7	137	0.31	0.81
	SD	3.5	7.1	9.3	10.61		
**ICV**	M	143	139.9	139.6	140.5	0.13	0.94
	SD	1.2	10.4	10.3	10.6		
**IRP**	M	134.8	135.1	135.8	137.4	0.13	0.94
	SD	4.8	9.7	10.1	6.9		
**IML**	M	116.5	117.6	113.7	112.4	0.42	0.74
	SD	7.6	14.8	9.6	14.6		
**IVE**	M	114	104.9	105	111.5	0.85	0.48
	SD	24	12	12.3	14.6		

Legend: FSIQ Full scale IQ, ICV = Verbal Comprehension Index; IRP = Perceptual Reasoning Index; IML = Working Memory Index; IVE = Processing Speed Index.

**Table 3 children-10-00042-t003:** Health-related problems.

		IQ Categories						Age Categories						
		Mildly Gifted	Moderately Gifted	Highly Gifted	Total	χ^2^	*p*	<6	6–7	8–10	11–14	Total	χ^2^	*p*
Allergies	No	10	20	4	34	0.43	0.80	3	15	11	5	34	4.1	0.25
	Yes	2	7	1	10			1	1	5	3	10		
Skin problems	No	8	22	4	34	1	0.59	3	12	14	5	34	2	0.57
	Yes	4	5	1	10			1	4	2	3	10		
Sleep problems	No	7	19	5	31	3	0.23	4	13	10	4	31	4.6	0.20
	Yes	5	8	0	13			0	3	6	4	13		
Food selectivity	No	6	15	2	23	0.44	0.80	2	11	7	3	23	2.9	0.40
	Yes	6	12	3	21			2	5	9	5	21		

**Table 4 children-10-00042-t004:** Developmental precocity.

		IQ Categories						Age Categories						
		Mildly Gifted	Moderately Gifted	Highly Gifted	Total	χ^2^	*p*	<6	6–7	8–10	11–14	Total	χ^2^	*p*
Language precocity	No	5	8	1	14	0.85	0.65	1	6	5	2	14	0.7	0.87
	Yes	7	18	4	29			3	9	11	6	29		
Motor precocity	No	12	22	4	38	2.3	0.32	4	14	14	6	38	2.3	0.51
	Yes	0	4	1	5			0	1	2	2	5		
Writing precocity	No	9	11	0	20	9.9	0.007	1	7	8	4	20	1.4	0.70
	Yes	2	15	5	22			3	9	6	4	22		
Mathematical precocity	No	9	8	2	19	7.7	0.02	0	5	11	3	19	10.6	0.01
	Yes	2	17	3	22			4	10	3	5	22		

Note: significant statistics are underlined.

**Table 5 children-10-00042-t005:** Children’s areas of potentiality.

			GIA	SAA	CT	VPA	ES	AA	SLS	H
			Yes	Yes	Yes	Yes	Yes	Yes	Yes	Yes
Mildly gifted	age	6–7	3	1	0	1	1	0	0	0
		8–10	5	1	1	0	2	0	0	0
		11–14	2	0	2	2	0	0	0	0
		Total	10	2	3	3	3	0	0	0
Moderately gifted	age	<6	4	0	2	0	0	0	0	0
		6–7	8	5	2	0	1	0	0	0
		8–10	5	1	2	4	2	0	0	2
		11–14	1	1	0	0	1	0	0	1
		Total	18	7	6	4	4	0	0	3
Highly gifted	age	6–7	1	0	1	0	0	0	0	0
		8–10	2	2	0	1	0	0	0	0
		11–14	2	0	0	1	1	1	1	0
		Total	5	2	1	2	1	1	1	0
Total	age	<6	4	0	2	0	0	0	0	0
		6–7	12	6	3	1	2	0	0	0
		8–10	12	4	3	5	4	0	0	2
		11–14	5	1	2	3	2	1	1	1
		Total	33	11	10	9	8	1	1	3

Legend: GIA = general intellectual abilities, SAA = specific academic aptitude, CT = creative thinking, VPA = visual and performing arts, ES = emotional skills, AA = athletic abilities, SLS = social and leadership skills, H = humor.

**Table 6 children-10-00042-t006:** Children’s socio-emotional characteristics.

			SFM	ES	SH	SJ	M
			Yes	Yes	Yes	Yes	Yes
Mildly gifted	age	6–7	3	1	1	0	0
		8–10	5	1	0	0	0
		11–14	2	2	0	0	0
		Total	10	4	1	0	0
Moderately gifted	age	<6	3	2	0	0	0
		6–7	11	2	0	0	0
		8–10	8	1	0	1	0
		11–14	3	0	0	0	1
		Total	25	5	0	1	1
Highly gifted	age	6–7	0	0	0	0	0
		8–10	2	1	0	0	0
		11–14	2	1	0	0	0
		Total	4	2	0	0	0
Total	age	<6	3	2	0	0	0
		6–7	14	3	1	0	0
		8–10	15	3	0	1	0
		11–14	7	3	0	0	1
		Total	39	11	1	1	1

Legend: SFM = stress and frustration management, ES = empathy/sensibility, SH = sensorial hypersensibility, SJ = sense of justice, M = mood.

**Table 7 children-10-00042-t007:** Parenting educational challenges.

			EM	SM	BM	SMA	CCC	SIM	UTM
			Yes	Yes	Yes	Yes	Yes	Yes	Yes
Mildly gifted	age	6–7	1	4	1	2	0	0	0
		8–10	2	4	3	0	0	0	0
		11–14	2	3	1	0	0	0	0
		Total	5	11	5	2	0	0	0
Moderately gifted	age	<6	2	0	1	1	1	0	0
		6–7	6	3	7	1	0	0	0
		8–10	4	3	2	1	2	0	0
		11–14	0	1	3	1	0	0	1
		Total	12	7	13	4	3	0	1
Highly gifted	age	6–7	0	0	0	0	0	0	0
		8–10	2	1	1	0	0	0	0
		11–14	2	1	0	1	0	0	0
		Total	4	2	1	1	0	0	0
Total	age	<6	2	0	1	1	1	0	0
		6–7	7	7	8	3	0	0	0
		8–10	8	8	6	1	2	0	0
		11–14	4	5	4	2	0	0	1
		Total	21	20	19	7	3	0	1

Legend: EM = emotional management, SM = school management, BM = behaviour management, SAM = sociality management, CCC = comprehension of child characteristics, SIM = Self-image, UTM = use of technology.

**Table 8 children-10-00042-t008:** Parenting strategies.

			ES	SI	SG	DS	SI	TMS	PA
			Yes	Yes	Yes	Yes	Yes	Yes	Yes
Mildly gifted	age	6–7	3	0	0	0	1	0	0
		8–10	3	4	2	1	0	0	1
		11–14	2	3	1	0	0	0	0
		Total	8	7	3	1	1	0	1
Moderately gifted	age	<6	3	0	0	0	0	0	0
		6–7	7	2	0	1	0	0	0
		8–10	4	3	1	0	0	0	0
		11–14	3	1	1	1	0	0	0
		Total	17	6	2	2	0	0	0
Highly gifted	age	6–7	0	0	0	0	0	0	0
		8–10	0	0	0	0	0	0	0
		11–14	2	0	0	0	0	0	0
		Total	2	0	0	0	0	0	0
Total	age	<6	3	0	0	0	0	0	0
		6–7	10	2	0	1	1	0	0
		8–10	7	7	3	1	0	0	1
		11–14	7	4	2	1	0	0	0
		Total	27	13	5	3	1	0	1

Legend: ES = educational strategy, SI = school intervention, SG = recourse to specialists in giftedness, DS = drive to sociality, SI = sustaining interests, TMS = time management strategies, PA = physical activity.

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
