# Peer review of "Gifted Children through the Eyes of Their Parents:Talents, Social-Emotional Challenges, and Educational Strategies from Preschool through Middle School"

_children, 2022, doi:10.3390/children10010042_

Round 1
Reviewer 1 Report
The study presented is complete, interesting and novel for the scientific field. The sample is not very large but it is sufficient to obtain results with scientific rigor and without doubt, the article is a solid basis for further studies on the same subject. The structure and general approach of the study is correct and the objective and results obtained are clearly understood. I would just like to make a small suggestion for improvement to the authors in relation to the bibliographic references included. Most of the bibliographic references are from many years ago; it would be advisable to add some more current ones so that recent results are given priority. However, this is simply a small suggestion that the authors should consider.
Author Response
Regarding updating bibliographic sources, we recognize that some citations are dated; unfortunately, the literature in this area is limited, and some of the relevant articles are many years old. Therefore, we are doing a final check on search engines to try to improve this aspect. Thank you for the valuable suggestion.
Reviewer 2 Report
The paper focuses on different aspects of the profile of gifted children that may influence parenting, health-related issues that may characterize gifted children, emotional peculiarities that parents identify in their children with respect to different ages and levels of giftedness, educational challenges and parental strategies at different age. The empirical material comes from a mixed methods study and the analysis in the paper is based on the results from ANOVA tests. My suggestions to the authors are the following. The quantitative analysis can be supported with quotes form the interviews since is the sample is small and perhaps the frequencies of some categories are low. The small sample size may influence also the statistical significance of the ANOVA tests and respectively, the conclusions drawn from it. Quotes and reflections from the in-depth interviews may complement and strengthen the findings from the statistical analysis and make more robust the conclusions draw from it. It is mentioned in the abstract that the evidence from the study can “sets the foundation for the development of parent support counselling programs”. However, the implications from the study are not commented in the manuscripts. Reflections on the practical implications with respect to the development of parent support counselling programs can be explained in a separate section in the manuscript.
Author Response
First of all, thank you for your valuable suggestions. First, we have supplemented the text (section 5.2 ANOVA) with excerpts from parent interviews; second, we edited the abstract and, finally, supplemented the conclusion with a short section on family support interventions.
We have also rechecked the English language.